

# Brief communication: Reduced bandwidth improves the depth limit of the radar coherence method for detecting ice crystal fabric asymmetry

Ole Zeising[1,2], Álvaro Arenas-Pingarrón[2], Alex M. Brisbourne[2], and Carlos Martín[2]

[1]Alfred-Wegener-Institut Helmholtz-Zentrum für Polar- und Meeresforschung, Bremerhaven, Germany
[2]British Antarctic Survey, Cambridge, UK

**Correspondence:** Ole Zeising (ole.zeising@awi.de)

**Abstract.** Ice crystal orientation fabric strongly affects the viscous deformation of glacier ice. A popular technique to investigate ice fabric is radar polarimetry, often analysed using the coherence method. However, in fast-flowing areas with strong anisotropy, this method provides information of shallow areas below the surface only. This study proposes reducing radar bandwidth to enhance the depth limit for fabric asymmetry detection. Using data from two ice streams, we demonstrate that reduced bandwidth significantly increases the depth limit, depending on the centre frequency. This approach aims to improve the understanding of the spatial distribution of fabric, crucial for ice dynamics in fast-flowing regions.

## 1 Introduction

The viscous deformation of glacier ice is controlled by its temperature and the bulk ice crystal orientation fabric (henceforth referred to as *fabric*). Due to the mechanical anisotropy of ice crystals, the influence of the fabric on the viscosity is directional: depending on the direction of deformation, the ice is softer or harder (Cuffey and Paterson, 2010). The orientation of the fabric depends on the past deformation of the ice and controls the present viscosity (Alley, 1988). However, little is known about the spatial and temporal distribution of the fabric, and, despite its importance, fabric is inadequately represented in large-scale ice flow models.

The most reliable method to determine the fabric is the analysis of ice cores (e.g. Weikusat et al., 2017). By analysing thin sections, the orientation of the crystallographic axis ($c$-axis) of each grain can be determined and represented by a second-order orientation tensor. The difference in the horizontal eigenvalues $\lambda_x$ and $\lambda_y$ of this tensor gives the horizontal fabric asymmetry:

$$\Delta\lambda_{xy} = \lambda_x - \lambda_y. \tag{1}$$

Because ice cores are rarely drilled in ice streams, little was known about the fabric orientation and strength of asymmetry in the fast-flowing areas of the Greenland and Antarctic ice sheets.

Since ice crystals exhibit both mechanical and dielectric anisotropy (birefringence), horizontal fabric asymmetry can also be estimated using polarimetric radar measurements (Hargreaves, 1978). Due to the dielectric anisotropy, the relative permittivity $\varepsilon$ and thus the propagation velocity of the electromagnetic wave depends on the orientation of the polarised wave. By conduct-



ing two perpendicular measurements with polarisations in $x'$ and $y'$ direction, the difference in permittivities $\Delta\varepsilon_{x'y'}$ can be determined, allowing for the calculation of horizontal fabric asymmetry using the relationship:

$$\Delta\lambda_{x'y'} = \frac{\Delta\varepsilon_{x'y'}}{\Delta\varepsilon'}, \tag{2}$$

with the maximum permittivity difference for ice-penetrating radar frequencies of $\Delta\varepsilon' = 0.034$ (Matsuoka et al., 1997).

A widely used method for determining the difference in permittivities is the coherence method (Dall, 2010). This involves measuring the phase delay from the coherence of two perpendicular co-polarised signals, $s_{hh}$ and $s_{vv}$, due to their propagation velocity differences. The coherence $c_{hhvv}$ is calculated over depth segments containing $N + 1$ bins, starting at bin $i_n$:

$$c_{hhvv} = \frac{\sum_{j=i_n}^{i_n+N} s_{hh,j} s_{vv,j}^*}{\sqrt{\sum_{j=i_n}^{i_n+N} |s_{hh,j}|^2} \sqrt{\sum_{j=i_n}^{i_n+N} |s_{vv,j}|^2}}, \tag{3}$$

where $*$ indicates the complex conjugate. The magnitude $|c_{hhvv}|$ indicates the degree of correlation and the argument $\phi_{hhvv} = \arg(c_{hhvv})$ gives the phase shift (range $-\pi$ to $\pi$). Its uncertainty can be estimated from the Cramer-Rao bound (Touzi et al., 1999; Jordan et al., 2019). From the depth gradient of the phase shift $\mathrm{d}\phi_{hhvv}/\mathrm{d}z$, the difference in permittivity can be calculated:

$$\Delta\varepsilon_{x'y'} = \frac{2c_0\sqrt{\bar{\varepsilon}}}{4\pi f_c} \frac{\mathrm{d}\phi_{hhvv}}{\mathrm{d}z}, \tag{4}$$

with the speed of light in vacuum $c_0$, the mean (polarization averaged) permittivity $\bar{\varepsilon}$, and the centre frequency $f_c$. While the coherence method was used successfully at ice divides and other slow flowing areas with weak anisotropy (Jordan et al., 2019; Young et al., 2021; Ershadi et al., 2022), it revealed the fabric asymmetry to only shallow depths in fast flowing areas with strong anisotropy, such as ice streams (Jordan et al., 2022; Zeising et al., 2023). Due to propagation velocity differences, signals with the same two-way travel time are actually the result of scatterers at different depths. When this difference in scatterer-depth exceeds the range-resolution of the radar system, coherence is lost (Leinss et al., 2016). To address this, the co-registration method was developed, which shifts the depth segment of one signal to maximise overlap (Zeising et al., 2023).

In this study, we propose reducing the frequency bandwidth of the radar system to enhance the depth range for determining fabric asymmetry with the coherence method. We test various bandwidths and frequency ranges using data from a widely used ground-penetrating radar system at the EastGRIP drill site, comparing our results with ice core analyses. We then apply our improved method to the Rutford Ice Stream, aiming to extend the depth range of fabric asymmetry determination. Finally, we describe the advantages and disadvantages of bandwidth reduction.

## 2 Methods

### 2.1 Coherence depth limitation

The coherence method is based on the coherence of the range-resolution cells of $s_{hh}$ and $s_{vv}$ with the same two-way travel time. The magnitude of the coherence decreases when the two cells are not perfectly overlapping in depth and thus do not



contain exactly the same scatterers. The overlap $1 - (\Delta r/\Delta R)$ is positive as long as the depth deviation of the two cells $\Delta r$ is smaller then the range-resolution $\Delta R$ of the radar system (Leinss et al., 2016). While the range-resolution depends on the frequency bandwidth of the radar system $B$

$$\Delta R = \frac{c_0}{2B\sqrt{\bar{\varepsilon}}}, \tag{5}$$

with the speed of light in vacuum $c_0$, and the polarization averaged permittivity $\bar{\varepsilon}$, the depth deviation is bandwidth independent:

$$\Delta r = z\frac{\sqrt{\varepsilon_{x'}} - \sqrt{\varepsilon_{y'}}}{\sqrt{\bar{\varepsilon}}}, \tag{6}$$

with the depth averaged permittivities $\varepsilon_{x'}$ and $\varepsilon_{y'}$ in $x'$ and $y'$ direction, and depth $z$.

The coherence method cannot be applied beyond the depth at which the depth deviation exceeds the range resolution, and there is no overlap in the range bins. The overlap is zero when the depth deviation equals the range-resolution, which gives the coherence depth limit as:

$$z = \frac{1}{2B}\frac{c_0}{\sqrt{\varepsilon_{x'}} - \sqrt{\varepsilon_{y'}}}. \tag{7}$$

Thus, the depth to which coherence can be calculated depends inversely on the difference in permittivity and on the radar bandwidth.

## 2.2 Phase-sensitive radar data analysis

We test the effect of a reduced bandwidth on two different polarimetric phase-sensitive radar data sets acquired with ApRES (Nicholls et al., 2015) in fast flowing ice:

1. Multi-polarised measurements at the EastGRIP (East Greenland Ice-core Project) drill site on the Northeast Greenland Ice Stream (Data: PpRES_CL, Zeising and Humbert, 2022; Zeising et al., 2023);

2. Quad-polarised measurements at the Rutford Ice Stream, Antarctica (Data: A3, Jordan et al., 2020, 2022).

The ApRES is a frequency-modulated continuous wave radar that is operated with a bandwidth of $200\,\mathrm{MHz}$ and a centre frequency of $300\,\mathrm{MHz}$. It transmits chirps of $1\,\mathrm{s}$ length while it increases the frequency linearly from $200\,\mathrm{MHz}$ to $400\,\mathrm{MHz}$. For a quad-polarised measurement, two co-polarised ($s_{hh}$ and $s_{vv}$) and two cross-polarised measurements ($s_{hv}$ and $s_{vh}$) were made. The multi-polarised measurements at EastGRIP consist of the four quad-polarised measurements with different azimuthal orientations. The benefit of quad- or multi-polarised measurements is that the full radar return from any antenna orientation can be synthesised from the co- and cross-polarised measurements using a matrix transformation (Ershadi et al., 2022).

For the specified bandwidth and frequency range, only the corresponding part of the chirp was used for preprocessing, which follows the methods described in Zeising et al. (2023). We compared the full $200\,\mathrm{MHz}$ bandwidth with limited bandwidths of $100$, $75$ and $50\,\mathrm{MHz}$, all applied at the respective minimum available centre frequencies (see Table 1).



**Table 1.** Bandwidth $B$, centre frequency $f_c$ and range-resolution $\Delta R$ used in the tests for the coherence method.

| Code | $B$ (MHz) | $f_c$ (MHz) | $\Delta R$ (m) |
|---|---|---|---|
| B200, fc300 | 200 | 300 | 0.42 |
| B100, fc250 | 100 | 250 | 0.84 |
| B75, fc237.5 | 75 | 237.5 | 1.13 |
| B50, fc225 | 50 | 225 | 1.69 |

After preprocessing, we synthesised $s_{hh}$ and $s_{vv}$ in 1° azimuthal intervals from 0 to 180°. For each azimuth, we calculated the coherence (Eq. 3), and hence the phase shift, with a moving depth segment of 20 m. We unwrapped the phase, smoothed it with a moving-average of 100 m and calculated its depth gradient with a moving window of 200 m. We then calculated the difference in permittivity and the horizontal fabric asymmetry.

We compare the results of the coherence method with the fabric asymmetry derived from the co-registration method (Zeising et al., 2023). Here, the full bandwidth of 200 MHz was used, since the co-registration method can be applied until the noise level depth limit, and it is thus independent of the coherence depth limit. At the EastGRIP site, we also performed a comparison of the coherence method with the fabric asymmetry derived from the EastGRIP ice core (Stoll et al., 2021; Weikusat et al., 2022). Based on the fabric asymmetry of the ice core, we calculated the coherence depth limit by using Eq. (2) to obtain the difference in permittivity from the fabric asymmetry. At the Rutford Ice Stream, the coherence depth limit was calculated assuming the permittivity difference derived with the co-registration method.

## 3 Results

### 3.1 Northeast Greenland Ice Stream

At the EastGRIP drill site, the coherence depth limit for the standard bandwidth of 200 MHz is about 380 m (Fig. 1a). With decreasing bandwidth the depth limit increases. For bandwidths of 100, 75 and 50 MHz, the coherence depth limit is 530, 625 and 810 m respectively. Thus, at the EastGRIP drill site, the bandwidth reduction from 200 to 50 MHz increases the depth limit of the coherence method by a factor of $\sim 2.1$.

The ice core derived horizontal fabric asymmetry at the EastGRIP drill site revealed a rapid increase from 0 to $\sim 0.5$ within the upper 100–500 m (Fig. 1b). Below these depths it increases only slightly up to 0.6 at a depth of 800 m. The same general distribution is shown by the co-registration method to beyond 1000 m depth. All horizontal fabric asymmetries estimated with the coherence method (for details see Figure A1) show the same rapid increase within the upper 350 m. At this depth, the fabric asymmetry for the full bandwidth of 200 MHz begins to differ from the ice-core derived asymmetry and starts to decrease rapidly. The asymmetry derived from the smaller bandwidth of 100 MHz matches the the ice-core derived asymmetry as well as the one from the co-registration method at least until a depth of 730 m, although the calculated depth limit is only 530 m. The same applies to the fabric asymmetry from the bandwidth of 75 MHz, but for roughly 100 m greater depth. However, the





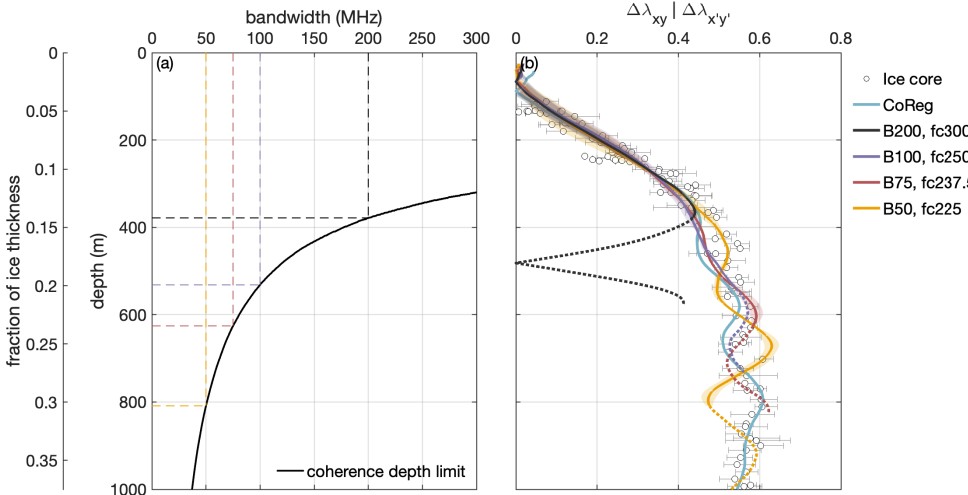

**Figure 1.** Comparison of horizontal fabric asymmetry $\Delta\lambda$ estimations. (a) Coherence depth limit of the coherence method as a function of bandwidth, calculated based on the permittivity difference derived from the fabric asymmetry of the EastGRIP ice core. Dashed lines show the depth limits for selected the bandwidths. (b) Fabric asymmetry determined from the coherence and co-registration method of polarimetric ApRES measurements at EastGRIP and from weighted horizontal eigenvalues from EastGRIP ice core (white dots) (Weikusat et al., 2022). The solid part of the coherence lines is above the depth limit, while the dotted part shows the first 200 m below that limit.

fabric asymmetry derived from ApRES data with 50 MHz bandwidth remains at a high level, similar to that from the other methods, but with larger deviations.

### 3.2 Rutford Ice Stream

At Rutford Ice Stream, a more detailed comparison of the coherence magnitude and phase shows a clear improvement for the
110 limited bandwidth of 75 MHz compared to the full bandwidth (Fig. 2). The coherence based on the limited bandwidth shows higher magnitudes (Fig. 2a,e) and clearer visible nodes (phase difference becomes an odd multiple of $\pi$ (Fujita et al., 2006)) in the wrapped phase (Fig. 2b,f) down to the depth at which the coherence depth limit is reached. Whilst the coherence depth limit is reached at 220 m for the standard bandwidth, it is twice this depth for the limited bandwidth case (440 m). Below these depth limits, the nodes fade out. The horizontal fabric asymmetry is based on the phase gradient of the unwrapped phase of a selected
azimuth (Fig. 2c,g). Both, the co-registration and the coherence methods show a rapid increase in horizontal fabric asymmetry to a value of 0.4. Whilst the fabric asymmetry from the co-registration method continues to increase to 0.6 at 300 m depth, the fabric asymmetry from the coherence method with standard bandwidth decreases (Fig. 2d). The asymmetry from the reduced bandwidth matches the fabric asymmetry of the co-registration method down to its calculated depth limit and beyond. Thus, applying the coherence method to reduced-bandwidth ApRES data as well as applying the co-registration method revealed the
strong fabric asymmetry and its depth profile at the Rutford Ice Stream beyond the shallow depths.



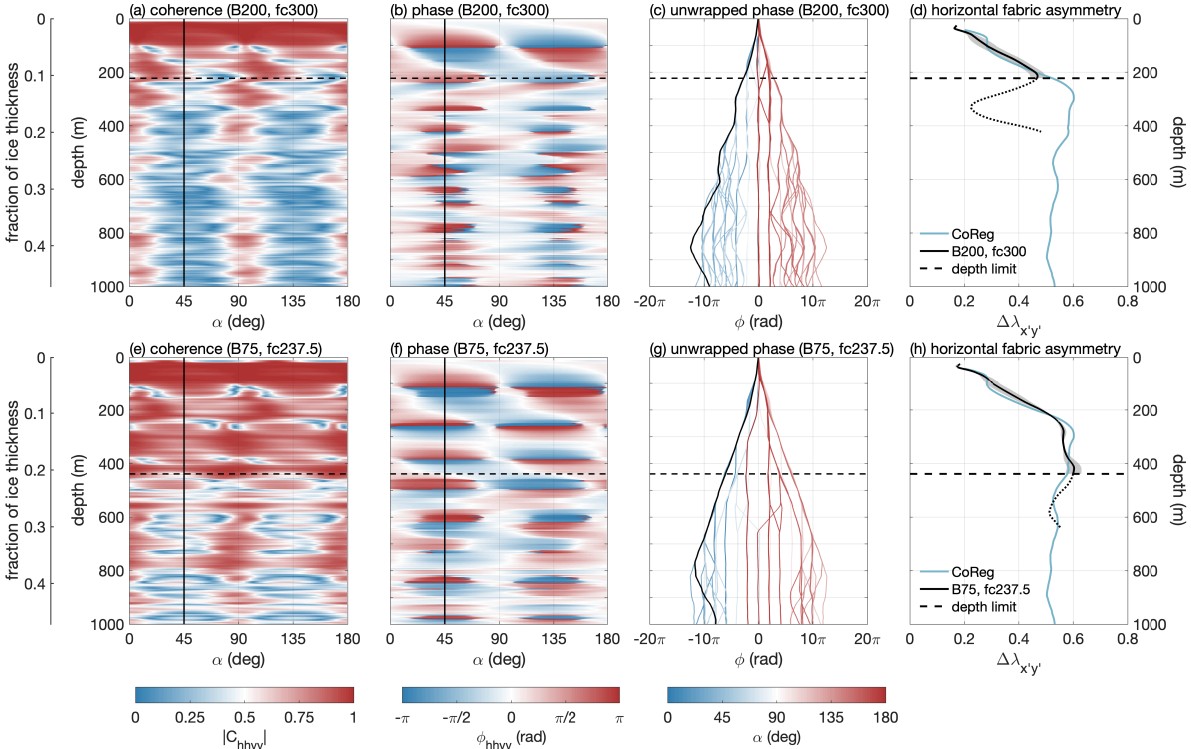

**Figure 2.** Coherence analysis of quad-polarised ApRES measurement at Rutford Ice Stream. (a–d) Analysis of the standard $200\,\mathrm{MHz}$ bandwidth ($f_c = 300\,\mathrm{MHz}$) and (e–h) of $75\,\mathrm{MHz}$ bandwidth ($f_c = 237.5\,\mathrm{MHz}$). The columns show respectively: (a,e) the magnitude of coherence for synthesised azimuths; (b,f) the wrapped phase shift; (c,g) the unwrapped phase shift; and (d,h) the derived horizontal fabric asymmetry of selected azimuths. The selected azimuths are marked by solid black lines (a–c, e–g). The horizontal dashed lines mark the coherence depth limit and the blue line shows the co-registration result.

## 4 Discussion

By evaluating the impact of bandwidth on the coherence method, we show that limited bandwidth allows the determination of the horizontal fabric asymmetry to a significantly greater depth. However, we find that the selected centre frequency also affects the quality of the results as suggested by radar theory. Additional tests, evaluating smaller bandwidths centred at higher

frequencies, do not indicate any improvement over greater bandwidths. Our preferred explanation for the influence of the central frequency is that higher frequencies, or lower wavelengths, provide a less accurate phase average within the range-resolution cell. With a larger range-resolution cell and a higher centre frequency, the phase variation within the cell increases and can exceed several phase cycles. This is because the phase variation is frequency-dependent, in contrast to the range resolution. The bandwidth and the centre frequency should therefore be selected to maximise the coherence depth limit for a suitable number

of phase cycles within the range-resolution cell.





Our tests of the bandwidth limitations of ApRES measurements shows that the best results were achieved with a bandwidths of 75 and 100 MHz for centre frequencies of 237.5 and 250 MHz, respectively. The phase variation within the range-resolution cell corresponds to $6.3\pi$, or $5\pi$ rad, respectively. At higher centre frequencies for the same bandwidths, the phase variation increases and led to a less accurate estimate of the horizontal fabric asymmetry (not shown). A further reduction of the band-
135 width to 50 or even 25 MHz, with minimum centre frequencies of 225 or 212.5 MHz, also led to a greater phase variation within the range-resolution of $9\pi$, or $17\pi$ rad, respectively. Again, the resulting fabric asymmetry from these settings did not lead to an improvement compared to the greater bandwidths. For these smaller bandwidths, the centre frequency would have to be significantly reduced, which is not possible with the ApRES system that uses a frequency range from 200 to 400 MHz. In addition to this explanation for the influence of the central frequency on the results, it is also possible that the antenna pattern
depends on the frequency range and thus affect the selection of a centre frequency. The pattern of the skeleton slot antennas that were used for the polarimetric ApRES measurements in this paper is unknown.

A reduced bandwidth is required to increase the overlap of the range-resolution cells in the coherence method. In the co-registration method (Zeising et al., 2023), the increase in overlap is achieved by shifting the cells. However, this requires a phase variation of less than $2\pi$ rad within the range-resolution cell, which is achieved by zero-padding (Brennan et al., 2014).
Our comparison of the coherence and the co-registration methods shows that the co-registration method is still preferable as it bypasses the depth limitation and provides higher resolution.

## 5 Conclusions

We show that the maximum depth at which ice fabric can be analysed using the coherence method on polarimetric radar data depends strongly on the frequency range used. We demonstrate that reducing both the bandwidth and centre frequency
improves the depth to which coherence can be determined, which is particularly important in areas of strong anisotropy such as ice streams. Our findings are supported by radar theory and field observations. Existing high-bandwidth data, such as those acquired with the widely used ApRES radar system, allow the reduction of the bandwidth at the data processing stage. Therefore, the method presented in this study can be applied to improve measurements of fabric asymmetry, and thus ice viscosity, using existing data. Knowledge of the spatial distribution of fabric asymmetry, especially over greater depths, will
be important in the future when large-scale ice flow models are able to account for the effects of fabric on viscosity.

*Data availability.* Raw data of the multi-polarised ApRES measurements at EastGRIP (https://doi.org/10.1594/PANGAEA.951267, Zeising and Humbert, 2022) and crystal $c$-axes of ice core samples are published at the World Data Center PANGAEA (https://doi.org/10.1594/PANGAEA.949248, Weikusat et al., 2022). Raw data of the quad-polarised ApRES measurements at Rutford Ice Stream are published at the NERC's Polar Data Centre (https://doi.org/10.5285/D5B7E5A1-B04D-48D8-A440-C010658EC146, Jordan et al., 2020).





**Appendix A:  EastGRIP coherence analysis**

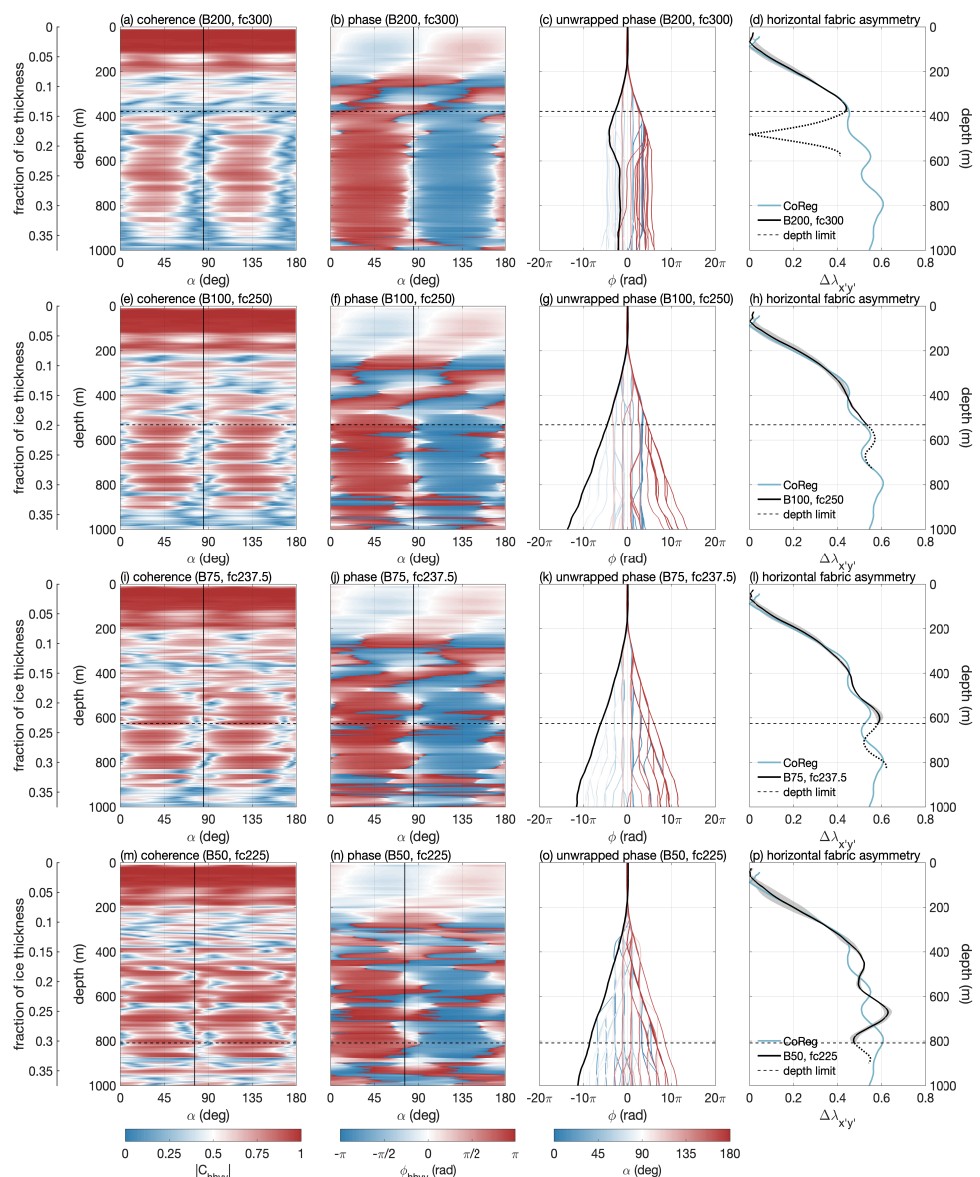

**Figure A1.** Coherence analysis of multi-polarised ApRES measurements at EastGRIP drill site. (a–d) Analysis of the standard $200\,\text{MHz}$ bandwidth ($f_c = 300\,\text{MHz}$); (e–h) bandwidth $100\,\text{MHz}$ ($f_c = 250\,\text{MHz}$); (i–l) bandwidth $50\,\text{MHz}$ ($f_c = 237.5\,\text{MHz}$); (m–p) bandwidth $50\,\text{MHz}$ ($f_c = 225\,\text{MHz}$). The columns show respectively: (a,e,i,m) the magnitude of coherence for synthesised azimuths; (b,f,i,n) the wrapped phase shift; (c,g,k,o) the unwrapped phase shift; and (d,h,i,p) the derived horizontal fabric asymmetry of selected azimuths. The selected azimuths are marked by solid black lines (a–c, e–g, i–k, m–o). The horizontal dashed lines mark the coherence depth limit and the blue line shows the co-registration result.



*Author contributions.* OZ and AAP developed the methodology with contributions from AB and CM. OZ wrote the manuscript that was reviewed by all authors.

*Competing interests.* Some authors are members of the editorial board of The Cryosphere.

*Acknowledgements.* EastGRIP is directed and organised by the Centre for Ice and Climate at the Niels Bohr Institute, University of Copenhagen. It is supported by funding agencies and institutions in Denmark (A. P. Møller Foundation, University of Copenhagen), USA (US National Science Foundation, Office of Polar Programs), Germany (Alfred Wegener Institute, Helmholtz Centre for Polar and Marine Research), Japan (National Institute of Polar Research and Arctic Challenge for Sustainability), Norway (University of Bergen and Trond Mohn Foundation), Switzerland (Swiss National Science Foundation), France (French Polar Institute Paul-Emile Victor, Institute for Geosciences and Environmental research), Canada (University of Manitoba) and China (Chinese Academy of Sciences and Beijing Normal University).

Funding:
The Rutford Ice Stream data were acquired as part of the Beamish project funded by NERC AFI award numbers NE/G014159/1 and NE/G013187/1. The work of OZ was funded by the German Academic Exchange Service (DAAD) Programme (Forschungsstipendien für promovierte Nachwuchswissenschaftlerinnen und -wissenschaftler; Kurzstipendien).



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
