# Peer review of "Brief communication: Reduced bandwidth improves the depth limit of the radar coherence method for detecting ice crystal fabric asymmetry"

_EGUsphere, 2024_

## Author Response (AR1)

Dear Reviewers,

We would like to thank you again for your efforts to improve our manuscript.

We followed the suggestions from both reviews and now provide an analysis of the centre frequency dependence of the coherence method for ApRES measurements in the revised version of the manuscript.

Thank you once again for your time and expertise.

Kind regards,

Ole Zeising and co-authors

**Authors point-to-point response on Referee Comment #1 to egusphere-2024-2519**

In this paper, the authors discuss the use of the coherence method on multi-polarization ice penetrating radar data as a tool to investigate properties of ice fabric. Specifically, they demonstrate that narrowing the radar signal's bandwidth helps provide information in deeper areas. They validate their approach from experimental data.
The manuscript is in the form of a brief communication. It is concise, well written, and easy to follow. The paper should be of interest of the EGUsphere readership. I only have the following comments:

1) On Page 3, line 80, the authors state "all applied at the respective minimum available center frequencies (Table 1)". I wonder, however, if it is better to keep the same center frequency and reduce the bandwidth of the signal. Rather than change both simultaneously. Can the authors provide a comparison of the results obtained when keeping the center frequency fixed? This would be useful to other radar systems, in which data is collected in ultra-wideband mode, and then sub-banded in post-processing.

Thanks for raising this point. In the revised version, we have updated Fig. 1 by dividing the results for a constant centre frequency (b) and for the corresponding minimum center frequency (c). In (b), we have included two fabric asymmetry lines for a bandwidth of 100 MHz and 75 MHz for a center frequency of 300 MHz. Both show that the results differ from the co-registration method and the ice core analysis, before the coherence depth limit is reached. The discussion now focuses on the frequency dependence of the coherence analysis.

2) Minor comments:

a) In equation (5), $c_0$ is defined as the speed of light in vacuum, but that variable is already defined for equation (4). Please consider eliminating redundant variable definitions

b) Page 3, ~line 64-65 "and on the radar bandwidth" consider changing to " and on the bandwidth of the radar signal"

We followed both of your suggestions. Thanks!

**Authors point-to-point response on Referee Comment #2 to egusphere-2024-2519**

This paper presents a method for increasing the depth at which polarimetric coherence between HH and VV radar measurements drops to zero due to lack of coregistration. The method simply reduces the bandwidth which coarsens (increases) the range resolution so that the two images stay co-registered for longer two-way travel time offsets. Since the HH/VV travel time offset increases with depth, this implies that the images will stay co-registered and coherent into greater depths. Since only a subband of the full bandwidth is used, the method allows a range of centre frequencies to be chosen. The authors report that the lowest centre frequency produces the best results reporting that higher frequencies resulted in "a less accurate estimate of the horizontal fabric asymmetry"; their preferred explanation is that the longer wavelength reduces the phase variations within a single range resolution cell. I think this explanation should be expanded in the following ways:

1. Please include plots as a function of center frequency along with an explanation of an evaluation metric.
2. I think higher frequencies should perform better as long as everything else (e.g. signal to noise ratio) is held constant. I suggest this because the predicted phase noise will be identical as a function of frequency under constant SNR, but the higher frequencies will cause the signal to generate greater phase variations that are therefore easier to see. In any case, I think a more thorough investigation is needed and recommend including:
   a. A short time Fourier transform (STFT) result with some multilooking/snapshot averaging of the HH and VV signals to estimate frequency dependence of the signal to noise ratio as a function of depth. (Regarding suggestion of STFT: I think any power spectral density estimation as a function of depth would probably be fine.)
   b. A plot of the coherence as a function of depth for several different frequencies some some subband (probably 75 MHz or whatever was producing the best results).
   c. Consider three other explanations that I think are slightly more specific than the one suggested at this point: 1) reduced SNR at higher frequencies; 2) depolarization of the HH or VV waves due to slight misalignments of the crystals would happen faster at higher frequencies/shorter wavelengths; and 3) the worsening of the specular layer assumption at higher frequencies in a way that increases decorrelation due to a difference in the HH and VV scattering from the layer roughness. Plots from (2a) and (2b) above may help support this argument and more specifically point towards the cause(s) of the reduced higher frequency performance.

Many thanks for your detailed suggestions on the frequency dependence of the coherence method. We agree that higher frequencies improve the sensitivity for detecting a phase variation. In the revised version, the discussion focuses now on the frequency dependence of the coherence method applied to polarimetric ApRES data. The analysis of the coherence and fabric as a function of centre frequency revealed errors in the phase unwrapping when the coherence is low, which is the case for higher centre-frequencies (new figure). We followed your suggestion and applied the STFT to the raw ApRES data to analyse the SNR as a function of depth and of the centre frequency. However, the estimated SNR exhibits no frequency dependence. Enhanced roughness of the radar layers may also have resulted in frequency-dependent decorrelation or depolarization, as previously suggested by Jordan et al. (2022).

Another reason for the observed reduced coherence might be the directivity of the antenna pattern within the ApRES bandwidth. The pattern has not been measured during ApRES deployment, and off-vertical sidelobes might collect clutter that adds to the wanted vertical returns. We expect that radar returns from nadir have greater propagation losses than clutter from off-vertical, for example due to losses caused by warmer ice. This could have more impact on higher frequencies, but we must measure the antenna pattern first. Unfortunately, within the scope of this study, we cannot provide a final answer to what has caused the reduced coherence for higher frequencies and can only speculate about the causes.

Also could the reference for equation (2) be more explicitly stated? I suggest adding (Zeising et al., 2023) just before the declaration of (2).

Thanks, we added "(Gerber et al., 2023)".

The paper is well written and I was able to prove all results presented from the descriptions and citations provided. The result is interesting, but given that coregistration is always possible and straightforward to apply, I think the most interesting result is the frequency dependence result that the authors refer to but do not show.

--

Two typographical corrections:

Our tests of the bandwidth limitations of ApRES measurements show that the best results were achieved with bandwidths of 75 and 100MHz for centre frequencies of 237.5 and 250MHz, respectively.

Thanks for finding these typos. Both are corrected.

**References:**

Gerber, T. A., Lilien, D. A., Rathmann, N. M., Franke, S., Young, T. J., Valero-Delgado, F., Ershadi, M. R., Drews, R., Zeising, O., Humbert, A., Stoll, N., Weikusat, I., Grinsted, A., Hvidberg, C. S., Jansen, D., Miller, H., Helm, V., Steinhage, D., O'Neill, C., Paden, J., Gogineni, S. P., Dahl-Jensen, D., and Eisen, O.: Crystal orientation fabric anisotropy causes directional hardening of the Northeast Greenland Ice Stream, Nature Communications, 14, 2653, https://doi.org/10.1038/s41467-023-38139-8, 2023.

Jordan, T. M., Martín, C., Brisbourne, A. M., Schroeder, D. M., and Smith, A. M.: Radar Characterization of Ice Crystal Orientation Fabric and Anisotropic Viscosity Within an Antarctic Ice Stream, Journal of Geophysical Research: Earth Surface, 127, e2022JF006 673, https://doi.org/10.1029/2022JF006673, 2022.